# Stigma experiences and adaptations in accessing healthcare services among hill tribes in Thailand: A qualitative study

Peeradone Srichan[1,2], Tawatchai Apidechkul [1,2]*, Ratipark Tamornpark[1,2], Thanatchaporn Mulikaburt[1], Pilasinee Wongnuch[1,2], Siwarak Kitchanapaibul[1,2], Panupong Upala[2], Chalitar Chomchoei[2], Fartima Yeemard[2], Anusorn Udplong[1], Onnalin Singkhorn[2,3]

1 School of Health Sciences, Mae Fah Luang University, Chiang Rai, Thailand, 2 Center of Excellence for Hill Tribe Health Research, Mae Fah Luang University, Chiang Rai, Thailand, 3 School of Nursing, Mae Fah Luang University, Chiang Rai, Thailand

* Tawatchai.api@mfu.ac.th

## Abstract

### Background

One of the significant barriers to accessing healthcare services is the stigma experienced from healthcare workers. Individuals can be significantly impacted by stigma owing to being classed according to particular characteristics, such as being tribal members. This study aimed to understand the experiences and adaptations of hill tribe people in Thailand, who face stigma when accessing healthcare services.

### Methods

A qualitative phenomenological method was used to elicit information from hill tribe members with prior experience accessing healthcare. A question guide was used to interview the participants. The interviews were conducted in private and confidential rooms in hill tribe villages in August 2021. Each interview lasted for 45 minutes.

### Results

A total of 85 people participated in the study: 25 men and 60 women. The Akha and Lahu people constituted the majority of the participants. Many had no education, and the average monthly income was 2,500 baht per family. Three forms of stigma were detected among hill tribe people accessing healthcare services in different hospitals: verbal stigma, physical stigma, and contempt. Three levels of impact were found: completely not understood with no effect, little understanding with little pain, and fully understood with full impact. Two reactions to stigma were identified: nonresponse and response (proper, nonproper immediate response and assertive response). Three factors were protective against stigma: speaking fluent Thai, wearing modern clothing, and the ability to pay medical fees.

**Data availability statement:** All relevant data are within the paper and its Supporting Information files.

**Funding:** We received the fund from The Center of Excellence for the Hill tribe Health Research, Mae Fah Luang University with the No. 1/2021. The funders had no role in study design, data collection, and analysis, the decision to publish, or preparation of the manuscript.

**Competing interests:** The authors have declared that no competing interests exist.

## Conclusions

Hill tribe people face several forms of stigma related to various levels of impact and different reaction approaches. Some factors can protect against encountering stigma while accessing healthcare services in hospitals. The implementation of programs to reduce stigma should focus on improving the understanding of people's different cultures and languages and effective communication skills for hill tribe people. The central government of Thailand should develop a national strategic plan to improve these socioeconomic statuses.

## Introduction

Stigma is a negative attitude that attacks individuals with specific characteristics, leading to mental, physical, or social suffering [1]. Goffman defined stigma as an attribute that is deeply discrediting [2]. Individuals living with lower economic [3–5] educational statuses [6] and with different cultures and social norms [7,8] are vulnerable to stigma. Many forms of stigma have been identified, including verbal [9] and physical [10] forms. In verbal stigma, voice and language are used by an individual to express a negative attitude toward the other and can present in various forms [11]. The impact can be significant and prominent when an individual has power over another [11,12]. Stigma is an issue when there is unequal access to fundamental human rights, including public health and medical care systems [13,14]. Both short- and long-term effects are commonly observed in accessing medical care by those who are oppressed by stigma. Stigma is an essential issue in almost all societies, in developed and developing countries, including Thailand, and is expressed in different patterns, forms, manifestations, and outcomes [15].

In 2022, Thailand implemented a universal health scheme to ensure that all people with a Thai identification (ID) card can access healthcare services by receiving free-of-charge essential medical care [16,17]. However, some groups of people, such as hill tribe people, have poor health status because of inadequate access to healthcare services [18]. Several health problems, such as hand-foot-mouth disease [19], among hill tribes have been reported to be disproportionate to those reported in the Thai population, and a large proportion of hill tribe pregnant women are not cared for properly, according to the national standard protocol [20]. A study examining the survival time of hill tribe people with HIV/AIDS reported that it was shorter than those in the general Thai population with HIV/AIDS [21]. A large proportion of hill tribe diabetes mellitus patients (DM patients) have a significantly lower rate of blood glucose control than Thai DM patients [22].

Additionally, the poor education and economics of hill tribe people were observed and could be sources of stigma [23–25]. The inability to communicate in Thai and not being granted a Thai identification card are significant drivers of stigma in the daily life of hill tribe people [26,27]. Stigma could play a role in reducing access to public health and medical services through governmental healthcare institutes. Qualitative studies in Thailand reported that some groups of people faced severe stigma in various settings, including within their own families and workplaces [28]. A few studies have reported on the stigma experienced and stigma adaptation among hill tribes while accessing health services at health institutes [20]. Accepting the situation and choosing a private clinic were their adaptations [20]. Stigma adaptation is the process of changing to learn and survive any individual suffering from stigma [29]. Therefore, this study aimed to understand the experiences of stigma and adaptations when hill tribes in northern Thailand access healthcare services. The findings can be used to

understand the phenomenon of stigma in accessing healthcare services among hill tribes and as sources of information for further healthcare service development.

## Methods

A qualitative phenomenological method was used to collect data from hill tribe people living in villages in the Mae Fah Luang and Mae Chan districts, Chiang Rai Province, Thailand. Four of 77 hill tribe villages in the district were selected as the study setting. The selected villages are located in border areas of Thailand and Myanmar. The hill tribe people who lived in the selected villages composed the study population. The study samples or participants were chosen from the selected villages that met the inclusion criteria. The inclusion criteria were hill tribe people aged ≥15 years who had attended a hospital at least twice within the previous year and who experienced stigma while accessing a service in a hospital.

A set of question guides was developed from a literature review and was tested with six key hill tribe informants (three women and three men) who had experienced stigma while accessing healthcare services. The questions were also assessed for validity by three experts: one hill tribe person who graduated with a university degree in health science, one hill tribe village headman, and one village health volunteer.

Nine questions were used as the question guide: (1) Have you ever experienced stigma while attending a hospital? (2) Could you please describe the stigma you experienced? (3) Who expressed stigma against you—a doctor, nurse, or someone else? (4) How frequently did you experience stigma? (5) How did you feel while experiencing stigma? (6) How did you respond to the event? (7) Did it impact your health? (8) Why did you not receive good service from the healthcare provider? (9) What were your expectations before accessing the healthcare service? (S1 File).

Village headmen were contacted after ethical approval was obtained from the Chiang Rai Provincial Public Health Office. Details of the study, including the objectives and inclusion criteria, were explained to them. Four out of 77 villages were selected on the basis of the following characteristics: one village close to a district hospital, one close to a primary healthcare hospital, and another far from any city or hospital. A systematic review [30] reported that 9–17 participants in in-depth interviews were common practice in a qualitative to obtain qualified findings. Therefore, at the beginning, 60 people with stigma experienced and met the criteria, and 15 participants from each village were screened and invited to participate in the study purposively.

The interviews were conducted between August and September 2021 in a private and confidential room in the community hall prepared for this purpose. The interviews were conducted by eleven researchers who were experienced in qualitative research, public health, epidemiology, medical anthropology, medical sociology, and psychology. All the interviewers were familiar with hill tribe people. Interviewers were trained in conducting in-depth interviews and had conducted at least three in-depth interviews in project work experience. Before the formal interviews, the interviewers introduced their background to the participants. The interviews lasted 45 minutes each in the Thai language. All interviews were recorded after approval was obtained from the participant, and field notes were taken. The audio records were typed and checked for errors before coding. The transcripts were returned to the participants to check the accuracy of the information before further analysis.

The saturation of information was discussed among the researchers while visiting villages several times to complete the interviews with 60 participants. After the visits, researchers always discussed and recorded the key points. Data saturation was reached when no more critical evidence, thoughts, experiences, or possible themes were detected during our discussions.

A total of 12 code names, definitions, and quotes were developed as the codebooks. The codes were used to extract the information from 85 transcripts by importing them into the NVivo program (NVivo, qualitative data analysis software; QSR International Pty Ltd., version 11, 2015) to derive the primary themes, including labels, elaborations, and illustrations. The final themes were developed from the components of the primary themes. The final themes were labeled, elaborated, and illustrated before the presentation. The researchers reconsidered the findings before conclusions were drawn.

### Ethics approval and consent to participate

All research protocols, including the question guide, were approved by the Chiang Rai Provincial Public Health Office (CRPPHO No. 69/2564). All activities were performed under the relevant guidelines and regulations. Before the interviews, all the participants were provided information. Informed consent was obtained voluntarily. Those selected participants who could not write were asked to provide a fingerprint representing informed consent voluntarily. For those younger than 18 years, informed consent was also obtained from their parents before the interviews. All audio records were properly destroyed after the completion of the analysis. Information from the study was presented anonymously without being able to identify an individual's details.

## Results

### General characteristics

A total of 85 participants were recruited for the study: 25 men and 60 women. Most were Lahu (36 people), Akha (34 people), and Thai Yai (11 people). The average age was 44.8 years (min=15, max=70). Sixty people (70.6%) were Christian, 71.85% had never attended school, and 59.8% worked in the agricultural sector, with a median monthly income of 2,500 baht ($84) per family (median=2,500, interquartile range (IQR)=1,000–3,750) (Table 1).
The analysis identified three types of stigma, three levels of impact, two reaction approaches, and three factors that protect hill tribe people against stigma while they access healthcare services in different hospitals. Most people preferred to access a primary healthcare hospital, a small hospital located at the village level. At this level, all primary care services are available, including family planning and reproductive health, immunizations for children, etc. The second choice for hill tribes is to access medical care at a district hospital, which is the secondary level—a few cases accessed a tertiary hospital in the city when they have a severe illness.

### Forms of stigma

Several forms of stigma that hill tribe people experienced, including verbal, physical, and contempt, particularly from healthcare workers working in secondary or district hospitals, were identified. Nurses were the major contributors to all forms of stigma against hill tribe people. The hill tribe people in Thailand experienced three different forms of stigma.

   **a. Verbal.**  Stigma through verbal language was the major form of stigma. The expression of healthcare workers through language greatly impacted these hill tribe members in a particular context of communication. A large proportion of the hill tribe members could not speak fluent Thai, including having difficulties with accents while speaking, which was not entirely understandable by native Thai speakers. This could lead to poor confidentiality and patients' fear of healthcare workers. This reflects the powerlessness of the patient in their interactions with the healthcare workers.

A 47-year-old woman said [P#55]:

> "When I visited a doctor due to a very painful stomach, a nurse asked me to cooperate with her in doing something. However, I had such pain in my stomach that I could not follow her. She

**Table 1. Characteristics of the participants.**

| Characteristics | n | % |
|---|---|---|
| **Total** | **85** | **100.0** |
| **Gender** | | |
| Male | 25 | 29.4 |
| Female | 60 | 70.6 |
| **Age** (years) | | |
| <35 | 22 | 25.9 |
| 36–50 | 32 | 37.6 |
| >50 | 31 | 36.5 |
| **Tribe** | | |
| Akha | 34 | 40.0 |
| Lahu | 36 | 42.4 |
| Thai Yai | 11 | 12.9 |
| Lisu | 3 | 3.5 |
| Yao | 1 | 1.2 |
| **Marital status** | | |
| Single | 11 | 12.9 |
| Married | 64 | 75.3 |
| Ever married/divorce/separated | 10 | 11.8 |
| **Religion** | | |
| Christian | 60 | 70.6 |
| Buddhist | 25 | 29.4 |
| **Education** | | |
| No education | 61 | 71.7 |
| Primary school | 6 | 7.1 |
| High school | 14 | 16.5 |
| College/University | 4 | 4.7 |
| **Occupation** | | |
| Agriculture | 49 | 57.6 |
| Unemployed | 18 | 21.2 |
| Employee | 12 | 14.2 |
| Merchant | 3 | 3.5 |
| Attending School | 3 | 3.5 |

then spoke badly to me. Even though I felt much pain, I also did not like what she said. However, I did not respond to her because I feared I might not receive good treatment from her."

A 65-year-old woman said [P#58]:

"Last month, I went to a hospital to get drugs, and I saw a nurse speaking harshly to an old Akha lady. She said something the old lady did not understand and spoke many blaming words. I really didn't like it".

A 47-year-old woman said [P#60]:

"Just last week, while I visited a doctor at a hospital, a nurse shouted at me. She was furious and said, "You are the highland people; why don't you understand what I say?" Indeed, I did not clearly understand what she said."

Being spoken to negatively by a nurse, particularly for those who are older and not fluent in Thai, is a pervasive stigma experienced among hill tribe people in Thailand. Negative verbal expressions had the most significant impact on the emotions of hill tribe people while they were attending a hospital. However, the patients did not respond to the healthcare workers because they feared that they would not receive good treatment.

**b.  Physical.**  Several physical actions by healthcare workers expressed stigma were reported by the participants, such as throwing paper, finger-pointing, not being spoken to and being ignored, or being the recipient of threatening actions. Almost all forms of stigma are made by healthcare workers working at district hospitals. These physical actions were reported by all of the participants who neither spoke nor understood Thai. This stigma was reported as most frequently occurring in outpatient and labor departments. Some women reported having been physically abused during labor.

A 16-year-old man said [P#25]:

> "Maybe 4–5 months ago, I got in a traffic accident and needed to have my wounds dressed at a hospital. The nurse who treated me processed me very roughly, which hurt. I saw that she treated me without standard care. I felt she was practicing wound dressing on me because she knew I was a hill tribe member. Before dressing my wound, she called my name and asked, "Are you a hill tribe?".

A 64-year-old man said [P#73]:

> "This was very important for my life. I went to see a doctor to get drugs for hypertension. I was diagnosed with hypertension several years ago. On that day, I left my home very early. I sat down close to a nurse when I reached the hospital. Immediately, the nurse moved to another place. I know that sometimes this happens, but in my case, I felt that the nurse did not like me and did not want to sit close to me. Yes, absolutely, I accepted it".

A 24-year-old man said [P#65]:

> "A couple of years ago, I had dengue fever, and unfortunately, I was admitted to a hospital. During admission, I received less care than another patient whose bed was beside mine at the hospital. The nurse had many conversations with that person but did not even look at me. I felt that I had a very high fever, and I needed someone to help me. I requested medicine from her, but she did not respond. I know she might have a lot of work to do, but she should care fairly for everyone the same."

Physical stigma is another essential type of stigma experienced by hill tribe people while they are attended to in hospitals. Physical stigma often occurs during medical procedures and in the documentary process.

**c.  Looks of contempt.**  In Thai culture, cynicism or a look of contempt is another stigma that significantly impacts the victim. Thai women are frequently subjected to this type of stigma, directed toward those who look inferior or are not equal in some aspect, particularly with respect to economic and educational status. Stigma expressed by a look of contempt can impact anyone. This form of stigma not only occurs between Thai healthcare workers and hill tribe patients but is also common in daily life among people with different economic or educational statuses.

A 53-year-old woman said [P#50]:

> "I clearly remember when I attended a hospital last month. I do not have an ID card and did not attend school. Therefore, I was not confident in talking about my symptoms with

the nurse because I was afraid that I would not speak well and make her misunderstand my points. However, all my words disappeared when I made eye contact with the nurse. My feeling at that time was that I was afraid to say anything to her because she looked at me so unfriendly".

A 47-year-old man said [P#54]:

"One day, I visited a doctor, and before meeting with th**em**, a nurse called me in to take my history. I felt like a tiny man when she looked at me harshly and said, "Do you have money to pay doctor's fees?" I spoke very little and said without confidence that I had little money when she looked at me and talked to me. Do you know it was very upsetting?"

A 49-year-old woman said [P#53]:

"This was a very particular experience to me. I remember two months ago when I visited a doctor, and a local Thai woman looked at me from head to toe. I felt that she did not like me. I could sense that she looked at me like a poor person. I did not like that situation."

Stigma, expressed as a look of contempt, impacts many people without verbal expression. Both healthcare workers and the general Thai people can easily inflict this form of stigma. Being looked at negatively can discourage people and make them see themselves negatively, especially hill tribe people who live in poor communities in Thailand.

## Level of impact

**a.  No understanding of the message, no impact.**  At this level, no impact was detected by the hill tribe individuals because they did not understand the meaning of the messages. The effect was detected by their relatives, who cared for them while at the hospital. Therefore, when the participants were asked about their experience of stigma, they responded that there was no impact; however, impacts were felt by their relatives, who took them to the hospital. This means that individuals who cannot speak Thai are not directly impacted by stigma, particularly when expressed verbally. Nevertheless, it affected their relatives, who fully understood the messages.

A 44-year-old woman said [P#51]:

"When I visited a doctor, I heard many words from the nurse but did not understand what she said. At that time, I felt nothing. However, after coming back home, my son told me that the nurse spoke to me very roughly and repeated it over and over! However, anyway, I do not know, and I do not want to know what she said."

A 60-year-old woman said [P#80]:

"A year ago, one of my nephews brought me to a hospital, and I was admitted. During the hospital stay, my nephew told me that a nurse said tough words to me, including saying that I had a bad smell and why didn't I take a shower?"

She added

"Luckily, I did not know what she said, and I felt nothing. Is that good?" (laughing).

A 44-year-old man said [P#49]:

> "I guess I heard many hard words from nurses while attending a hospital. However, I did not understand Thai, so I did not know the meaning. However, my son said that the nurses were rough to me. However, I did not feel bad anymore".

**b. Little understanding of the message, little pain received.** Some hill tribe people spoke Thai and understood the meaning of what the healthcare worker said, making them feel bad. Most conflict occurred when the patient tried to explain what he or she wanted or needed help with, but the healthcare worker did not wholly understand the main ideas. The level of the response from the healthcare workers to the patients was not proper or adequate or did not directly respond to the need. Without effective communication, misunderstandings can impact several things, such as being misdiagnosed and misprescribed.

A 65-year-old woman said [P#58]:

> "Basically, I can use Thai a little. One day last year, I went to see a doctor for strong abdominal pain. I told a nurse about my problem, but she did not understand what I said. Moreover, I knew that she complained about me, but I did not clearly understand. The next day, my son told me that the nurse said tough words to me, but I felt nothing because I did not clearly understand what she said."

A 47-year-old woman said [P#57]:

> "Last year, I met a nurse while attending a hospital to get medicine for my hypertension illness. She told me something very fast in Thai. After a while she asked me, "Did you understand what I said?" I nodded my head to show her that I understood. In fact, I received little information from her. Do you know why I did that—because I was afraid she would not care for me properly."

**c. Fully understood message, full impact received.** Stigma can cause harm and pain to anyone, and the severity of the problem increases when a completely negative message is communicated from one person to another. Hill tribe people, who fully understood the negative messages from healthcare workers, experienced hurt and pain. Almost all hill tribe people under 30 years of age speak fluent Thai, and this particular group is impacted by a full understanding of the negative messages of healthcare workers.

A 16-year-old woman said [P#25]:

> "Last year, I was referred to an emergency room due to a car accident. It was almost 11:30 pm. The nurse did the wound dressing very roughly, which hurt a lot. During this time, she swore that the hill tribe people never used motorcycles carefully, which is not true. The accident was caused by heavy rain. I felt she liked this because we are the hill tribe."

A 28-year-old woman said: [P#23]:

> "Last week, I visited a hospital. One of the staff members asked me, "Do you have a Thai ID card?" I felt that this question had the same meaning as "Do you have money?" She asked me other questions without looking at me, such as "Which mountains are you from?" I felt terrible and shy."

A 30-year-old man said [P#42]:

> "Two years ago, I brought my father to the hospital. I heard a nurse ask questions to a woman close to me who was a hill tribe member and could not understand Thai. The nurse said, "Most hill tribe people do not understand anything." I looked at her and felt very angry even though she did not talk with me directly."

The impact of stigma depended on how much a person can use and understand Thai. The stigma had little impact on those who understood only a small amount of information because they could not understand the meaning of the messages. The fear of being improperly cared for was another concern among the hill tribe people, who faced stigma from healthcare providers.

## Reactions to experiences

Several reactions were observed by hill tribe people when they experienced stigma from healthcare workers. Nonresponse was one of the forms of ignorance to undesirable events experienced by the patient. This did not necessarily mean that they did not feel suffering but that they felt they had no choice but to be deeply hurt. Nonresponse indicated inequity. Some people remained silent because they feared that they would not receive good care from healthcare workers. Other people choose to respond in proper or improper ways, including aggressively.

   **a.  Nonresponse.**  There were two reasons for nonresponse to stigma. First, the recipient understood everything but kept silent out of fear. They feared they would not receive good-quality care from healthcare workers. This is the result of unequal power, especially concerning the ability of power to save life. The second reason was that patients did not understand the context of what was said in Thai. Therefore, they did not respond to the situation. Nonresponses due to a lack of understanding of the context commonly occur among elderly people. Owing to the limited use of Thai and different cultures, they did not understand basic aggressive words and negative physical language.

A 35-year-old woman said [P#33]:

> "A year ago, I brought my mom to visit a hospital. A nurse spoke to my mom and me harshly, and she blamed us poor people who could not afford to pay for the treatment. However, we did not respond to her. We kept quiet because we feared she might not care for us properly."

A 47-year-old woman said [P#55]:

> "Five years ago, I got a hand injury while working on the farm. I went to see a doctor at a hospital. Before meeting the doctor, I had been asked many questions by a nurse in the emergency room. Some questions hurt me, but I did not respond because I feared I would not be cared for properly."

A 42-year-old woman said [P#56]:

> "Last year, I went to see a doctor about my diabetes mellitus. One of the staff told me to fill out a form. I told him that I could not write. Immediately, he shouted at me with some bad words. I was sitting close to him, and I did not know what to do next. I kept everything in my heart because I feared he would not care for me."

**b. Response.** Responses to bad situations, including conversations and other interactions between hill tribe patients and healthcare workers, were presented at three levels: proper, nonproper immediate, and aggressive responses. The responses of hill tribe members who were fluent in Thai and aged 35 years and younger were recorded. In this age group, almost all the participants had attended school at some level and were exposed to people outside their village, including those who were familiar with other cultures and lifestyles. Because they were frequently exposed to negative messages and actions, these patients preferred to respond when they experienced negative comments or improper actions from healthcare workers.

Proper response: In this type of response, the patient addresses the problems calmly and directly when they arise. Although most responses were initiated by relatives, this indicates that patients often hold less power in interactions with healthcare workers.

A 23-year-old man said [P#9]:

> "I remember that one day in 2017, I visited a hospital with a severe headache. A nurse called my name, asked me for my history, and checked my blood pressure. The nurse asked me to explain my problem. I had only said a few words when she shouted, "Go wait there." I kept everything normal and responded, "I would very much appreciate it if you told me with normal words. I understand how busy you are today." I smiled at her."

A 30-year-old man said [P#10]:

> "Last year, I brought my wife to see a doctor at a primary care hospital to check her health during pregnancy; this was a small health center in our village. The staff member asked my wife some questions, and I responded that she could not speak Thai. The staff member turned to my wife and asked why she could not speak Thai. I was slightly surprised and responded, "How many languages do you speak because of graduating with a university degree? She never attended school—why do you expect her to speak Thai?"

Nonproper immediate response: Patients were the primary responders in this response type. When hill tribe patients suffer from illness, they responded nonproperly and immediately to healthcare workers who expressed any kind of stigma.

A 47-year-old man said [P#54]:

> "Last month, when I visited the diabetes mellitus clinic at a hospital, the staff member shouted at me with hard words. I looked at her face immediately and then stood up. Afterward, she calmed down and talked to me normally."

Aggressive response: Some people reported aggressive responses. A long wait time or ignoring severe suffering were sources of aggressive reactions. Aggressive responses were frequently reported in outpatient departments and labor rooms when patients were suffering severely but did not receive proper care.

A 20-year-old woman said [P#76]:

> "Last year, my son, who was five years old, got a foot injury, and I brought him to a hospital emergency room. A nurse cleaned his wound very roughly, and my son told me that it hurt a lot. I felt that she did not treat him carefully and mindfully. I immediately told her, "Could you please treat my son better, as he is a human? I felt very angry."

A 65-year-old man said [P#65]:

> "One day, a couple of weeks ago, I visited a hospital because I had a very severe headache. I waited in a long queue. A few moments later, a man dressed in a very modern style came and jumped the queue in front of me, and the nurse said it was fine. I felt it was not okay, so I asked the nurse to explain what happened. I started to ask many people who were sitting down in that area."

Many hill tribe people chose not to respond to the stigma because they feared being improperly cared for. When people did respond, they did so using three forms of the response: proper, nonproper, and aggressive. The responses depended on the situation and whether the patient was a loved one, such as a son.

## Protectors

While hill tribe people commonly encounter stigma, at least three factors act as protective factors against stigma in the context of accessing healthcare services: speaking fluent Thai, wearing modern clothing, and having money to pay for services.

a. *Speaking fluent Thai.* Among hill tribe people, speaking fluent Thai protected against stigma in any form received from healthcare workers. The ability to communicate smoothly and effectively in Thai reduced the incidence of stigma and its adverse effects. The fluent use of Thai increased the respect of healthcare workers.

A 26-year-old man said [P#82]:

> "I am an Akha, and I graduated high school. I can speak Thai very well. I know some people look physically different, which cannot be changed. However, we can adapt ourselves. When I go to a hospital, I can speak Thai and never have a problem with them. Speaking Thai truly helps me."

A 24-year-old man said [P#63]:

> "A couple of years ago, I visited a hospital to seek care after getting in a traffic accident. In the emergency room, I received good treatment. The nurse was very polite to me. This was because I could speak with them very well, even though I am Akha."

A 49-year-old woman said [P#84]:

> "I am Lahu, and I have lived in Thailand for 20 years. I did not attend school but can speak Thai because of my daily practice with Thais. I have not had any problems while visiting hospitals."

b. *Wearing modern clothing.* Hill tribe people who wear traditional clothes are more likely to experience stigma or negative experiences with healthcare workers. When hill tribe people wore modern clothing, they experienced less stigma. Stigma could be completely absent if they wore modern clothes and spoke fluent Thai. However, most hill tribe women in all tribes commonly wear their traditional style of clothing, which is part of their social culture. Thus, hill tribe women face more frequent and severe stigma when accessing health services. Women who wore traditional clothing and could not speak fluent Thai faced more severe stigma from healthcare workers.

A 47-year-old woman said [P#57]:

> "Actually, the hill tribe people always dress in the traditional form. However, I prefer to dress in modern clothes when visiting a doctor in a hospital. I feel that I receive more respect from people, including healthcare staff, while wearing modern clothes."

A 20-year-old woman said [P#61]:

> "I am Akha. I have noticed unfriendly looks from people around me. I decided to wear modern clothes, and I was surprised that I received less negative attention from people around me when I went to the city and hospital."

c. *Affordability for medical fees.* Most hill tribe people who can afford medical fees prefer to visit private hospitals. In a private hospital, services are based on a mix of standard health professional practices and private standard services. Anyone who accesses a private hospital will be served better than at public hospitals, which are nonprofit organizations. Hill tribe members who can afford it choose private hospitals to reduce the incidents of stigma. In addition, the service is always better and quicker than at public hospitals.

A 43-year-old woman said [P#39]:

> "Three years ago, I felt terrible while visiting a hospital. I was asked for money to pay medical fees before receiving care. I told them that I had a lot of money, and the actions of the hospital staff changed immediately. They were nice to me, but it's not right!"

A 28-year-old woman said [P#23]:

> "I prefer to go to a private hospital. Although we have to pay medical fees, we will receive good care. In a public hospital, I do not need to pay all the medical fees because I have a Thai ID card, but I feel that I have not received proper care from them. My friends say that money is very important to obtain good service from a hospital."

Those who dressed in modern clothing, spoke Thai well, and could afford to pay medical fees experienced less stigma. These characteristics indicate a higher level of socioeconomic status in the social environment. Those who had a lower socioeconomic status suffered in two ways—from their lower status and from experiencing stigma.

Domination by a majority produces significant stigma, stimulating inequity in accessing healthcare services by minority people through discounting the value of different cultures and socioeconomic groups. The ability of minority and vulnerable populations, owing to their socioeconomic status, to make a difference by setting their profiles as the standard for humankind is a source of both intentional and unintentional stigma.

## Discussion

The hill tribes living in northern Thailand suffer from several forms of stigma when accessing healthcare services, including verbal and physical stigma and looks of contempt. Different levels of impact from stigma were identified depending on the understanding of Thai. Individuals who did not understand the messages did not experience the effect of stigma. Those with little understanding of the message experienced little pain, whereas those who fully understood the message were extremely affected. Nonresponse and response were the two forms of

reaction to the stigma. Speaking fluent Thai, wearing modern clothing, and the ability to pay medical fees were found to be protective against stigma.

A large proportion of the hill tribe population has no formal education, leading to poor health literacy, which may result in limited access to healthcare services. The geographic isolation of their residency and traditional farming could impact their children's ability to access education and be a barrier to accessing health services for themselves and their children's generations [31]. Some basic profiles of hill tribe people seem to act as complementary factors of exposure to stigma.

The language barrier was detected as the major influencer of stigma expressed by Thai healthcare workers toward hill tribe patients. Several dimensions and levels were detected as the impacts of stigma in different situations. Surprisingly, the ability to understand the message could lead to varying effects on the experience of stigma. Understanding a message in a particular context of communication could vary its impact on the message receivers. A study confirmed that verbal stigma could cause suffering to individuals and reduce their self-esteem, which could have several consequences throughout their lifespan, especially for those in vulnerable populations [32]. In our study population, the major cause of verbal stigma was the insufficient understanding of Thai. During the interactions between the hill tribe patients and healthcare workers, particularly while the medical treatment procedures required full cooperation from patients, the patients did not understand the messages. These situations could lead to the development of intentional and unintentional actions, which are sources of stigma [33].

Physical stigma, including looks of contempt, expresses a sense of looking down on another person, which was found to be a form of stigma. Several studies [34–36] reported that being stigmatized by healthcare workers was strongly associated with poor health and healthcare quality, especially for those living with chronic diseases. A report from Thailand demonstrated that stigma was a major cause of poor treatment outcomes among HIV/AIDS patients and that promoting a good organizational culture and developing healthcare workers' interpersonal skills could minimize the impact of stigma on patients [37]. Thus, any form of stigma can negatively impact the treatment outcomes of patients. In some cultures, some form of stigma, such as looks of contempt, can deeply hurt the receiver or the victim.

Different levels of socioeconomic status were identified as significant sources of stigma while accessing healthcare services. A study conducted in Indonesia reported that patients who had a lower socioeconomic status and severe diseases were more affected by stigma than those with better health status [38]. The stigma experienced while accessing healthcare services among hill tribes in Thailand is a significant issue that must be considered and improved.

Owing to the specific characteristics of hill tribe people in Thailand, some prefer not to respond to avoid receiving poor care. Others responded with either a proper or nonproper approach, including aggressive responses to the situation. With repeated experiences of stigma, some hill tribe people prefer to respond improperly or aggressively. After a literature search of several data sources, no reports were identified that investigated responses to stigma, particularly in the healthcare setting. Owing to long-term experiences of stigma, as well as suffering from medical conditions, some patients prefer to respond. Many hill tribe members preferred to limit their response because they feared not being appropriately treated by healthcare workers. An aggressive response was reported by a patient fluent in Thai who was upset by the negative words of a healthcare worker and who was socioeconomically equal to the healthcare workers. This finding indicates that if hill tribe people have a high socioeconomic status, they may voice their opinions more often.

In our study, three characteristics of hill tribe people were identified as protective factors against stigma from healthcare workers: the ability to speak Thai fluently, wearing modern

clothing, and having money to pay for medical fees when attending a hospital. However, some of these characteristics are difficult to improve or adopt, mainly speaking Thai fluently and having money to pay for medical fees. The researchers observed that adaptability decreased among hill tribe members over 40 years of age because they had never attended school and typically lived in poor families. However, people younger than 30 years have more opportunities to reduce stigma and its impact because they attend school, can speak fluent Thai, and often have better jobs. Although no previous studies support our findings within the specific context of this population, our findings indicate that people can adapt to avoid or reduce the impact of stigma in their lives. To address the stigma among hill tribe people, studies should focus on the differences in stigma experienced in accessing different clinics. Further studies should also focus on stigma experiences in different tribes and age categories, including healthcare providers' perspectives on the issue. The future findings could lead to the need for different interventions to address these findings.

## Conclusion

Three forms of stigma are detected in the particular context of hill tribe people accessing healthcare services in different hospitals: verbal stigma, physical stigma, and a look of contempt. Those who had no understanding of the comments from the healthcare provider were not affected by the speech. In contrast, if they have some understanding of Thai, they will suffer somewhat, and if they fully understand the comments, they will experience the full impact of the stigma. Two reaction approaches are identified: nonresponse and response (proper, nonproper immediate response and aggressive response). Three factors are found to be protective against stigma: speaking fluent Thai, wearing modern clothing, and the ability to pay medical fees.

All healthcare workers who are working or who are assigned to work in hill tribe villages should be trained in specific programs to improve their knowledge and attitudes when working with people from different backgrounds. The provincial public health office should develop a monitoring system to ensure that everyone is cared for without stigma. Training healthcare workers assigned to work in hill tribe villages in basic hill tribe languages is also strongly recommended to avoid the stigma of language barriers. Putting effective interventions to induce a sense of equity in access to healthcare services without the matter of socioeconomic and cultural differences should be an essential element for training healthcare providers.

## Supporting information

**S1 File. Question guide.**
(DOCX)

## Acknowledgments

We are also grateful to all village headmen who worked in the study setting for their assistance in obtaining access and recruiting participants. Finally, we thank all the participants for participating in the study.

## Author contributions

**Conceptualization:** Peeradone Srichan, Tawatchai Apidechkul.

**Data curation:** Peeradone Srichan, Tawatchai Apidechkul, Ratipark Tamornpark, Thanatchaporn Mulikaburt, Pilasinee Wongnuch, Siwarak Kitchanapaibul, Panupong Upala, Chalitar Chomchoei, Fartima Yeemard, Anusorn Udplong, Onnalin Singkhorn.

**Formal analysis:** Peeradone Srichan, Tawatchai Apidechkul, Ratipark Tamornpark, Thanatchaporn Mulikaburt, Pilasinee Wongnuch, Siwarak Kitchanapaibul, Panupong Upala, Chalitar Chomchoei, Fartima Yeemard, Anusorn Udplong, Onnalin Singkhorn.

**Funding acquisition:** Tawatchai Apidechkul.

**Investigation:** Peeradone Srichan, Tawatchai Apidechkul, Ratipark Tamornpark, Thanatchaporn Mulikaburt, Pilasinee Wongnuch, Siwarak Kitchanapaibul, Panupong Upala, Chalitar Chomchoei, Fartima Yeemard, Anusorn Udplong, Onnalin Singkhorn.

**Project administration:** Tawatchai Apidechkul.

**Writing – original draft:** Peeradone Srichan, Tawatchai Apidechkul, Ratipark Tamornpark, Thanatchaporn Mulikaburt, Pilasinee Wongnuch, Siwarak Kitchanapaibul, Panupong Upala, Chalitar Chomchoei, Fartima Yeemard, Anusorn Udplong, Onnalin Singkhorn.

**Writing – review & editing:** Peeradone Srichan, Tawatchai Apidechkul, Ratipark Tamornpark, Thanatchaporn Mulikaburt, Pilasinee Wongnuch, Siwarak Kitchanapaibul, Panupong Upala, Chalitar Chomchoei, Fartima Yeemard, Anusorn Udplong, Onnalin Singkhorn.

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
