## [Decision Letter · Decision Letter 0]

12 Jul 2023

PONE-D-23-10436Stigma experiences and adaptations while accessing health care service among hill tribes in Thailand: a qualitative studyPLOS ONE

Dear Dr. Apidechkul,

Thank you for submitting your manuscript to PLOS ONE. After careful consideration, we feel that it has merit but does not fully meet PLOS ONE’s publication criteria as it currently stands. Therefore, we invite you to submit a revised version of the manuscript that addresses the points raised during the review process.

In submitting your revised manuscripts kindly pay close attention to the following areas of concern: 1. Substantially revise your introduction to provide the necessary background for your study, including clarification on different types of stigma and clarify distinctions between stigma from discrimination, supported by appropriate references.2. Please revise your "materials and methods" section to clarify your sampling frame, as well as inclusion and exclusion criteria. Also clarify issues of participant selection and methods applied to minimize bias in your study.3. Please revise your "results" section to clarify major themes and subthemes. Also describe how data saturation was achieved, and revise Table 1 as recommended by reviewers to  exclude/correct inappropriate columns and unnecessary inclusion within your Tables.4. Revise your discussion section with comparative analysis to similar studies in the literature, with appropriate referencing and how these other studies compare with findings from your own study.5. Clarify your Conclusions as well as your recommendations.6. In addition, substantially revise your entire manuscript to address all grammatical and typographical errors. Consultation with an English language editing service is recommended to address this issue if necessary or if required by the authors.7. Kindly correct and edit to your COREQ checklist to include page numbers with reference to the manuscript.

8. Please address all other relevant comments as annotated on the manuscript by reviewer #1 where necessary.

We look forward to receiving your revised manuscript.

Kind regards,

Sylvester Chidi Chima, M.D., L.L.M., LLD.

Academic Editor

PLOS ONE

Journal Requirements:

"We thank Mae Fah Luang University and The Center of Excellence for The Hill Tribe Health Research for supporting the grants. "

Additional Editor Comments: See marked up copy of manuscript with detailed comments by Reviewer #1.

Reviewers' comments:

Reviewer's Responses to Questions

**Comments to the Author**

1. Is the manuscript technically sound, and do the data support the conclusions?

Reviewer #1: Partly

Reviewer #2: Yes

2. Has the statistical analysis been performed appropriately and rigorously?

Reviewer #1: N/A

Reviewer #2: N/A

3. Have the authors made all data underlying the findings in their manuscript fully available?

Reviewer #1: Yes

Reviewer #2: Yes

4. Is the manuscript presented in an intelligible fashion and written in standard English?

Reviewer #1: No

Reviewer #2: Yes

5. Review Comments to the Author

Reviewer #1: This topic would add knowledge to stigma-related studies. Nevertheless, the manuscript needs tremendous flow, coherency, and sentence structure improvement. The current writing style needs to be more scientific. A professional English editing service is deemed necessary.

The introduction did not clearly justify the rationale on the conduct of this study.

The method need to be explained explicitly.

The discussion is not adequate.

The recommendation in the conclusion appeared abruptly.

Reviewer #2: Very great effort in bringing this paper, and paper looks wrote with the fashion and interest.

Some comments,

1. Stigma and discrimination is two different concepts with thin line difference like stigma is more with thought process and discrimination is behaviour or action. Most of your results section falls in discrimination aspects than limiting to stigma hence relook into the concepts once again.

2. Difference of information on participant selection: earlier authors mentioned person who experienced stigma has been interviewed (How did you screened them before selection please describe), then you mention inclusions, people who attended health care at least once or twice in last year was recruited so little this section is little confusing so think on it.

3. You mentioned about the participants selection process however some methodological perspectives missing in terms of sample, sampling techniques etc.

4. You have collected great response participants however interview guide looks very simple and having many closed ended question so justify how you were able to extract more response from participants

5. Please describe the inclusion criteria.

6. Author mentioned reinterview was carried out- There would be possibility of bias hence justify this section

7. Data saturation: provide details on How many researcher and their expertise

8. Small error in table one entry please cross check- Misplaced with row

9. In result a. verbal theme first quote looks more of language barrier than stigma so relook into it

10.In conclusion, author had mentioned about three forms of stigma which may confuse the reader since we have other types of stigma and discrimination like self, family, public, structural hence i recommend to rewrite this line or statement.

6. PLOS authors have the option to publish the peer review history of their article (what does this mean? ). If published, this will include your full peer review and any attached files.

**Do you want your identity to be public for this peer review?** For information about this choice, including consent withdrawal, please see our Privacy Policy .

Reviewer #1: No

Reviewer #2: No

---

## [Author Response · Author response to Decision Letter 1]

16 Jan 2024

Response to reviewers

Dear Editor,

Thank you very much for allowing us to revise this paper with great comments from reviewers. We are also sorry for the delay in the submission of our revised version because the corresponding author was detected by COVID-19 twice while revising the manuscript, including moving to the United States to study at Harvard Medical School, please accept our apologies for the delay in resubmitting the manuscript. After re-analysis of the data, we seriously revised and imrpoved the whole manuscript, particularly the introduction, methods, and discussion sections including change references. We do very much hope that you are happy with this version.

Thank you,

TK

Editor’s comments

Thank you for submitting your manuscript to PLOS ONE. After careful consideration, we feel that it has merit but does not fully meet PLOS ONE’s publication criteria as it currently stands. Therefore, we invite you to submit a revised version of the manuscript that addresses the points raised during the review process.

In submitting your revised manuscripts kindly pay close attention to the following areas of concern:

1. Substantially revise your introduction to provide the necessary background for your study, including clarification on different types of stigma and clarify distinctions between stigma from discrimination, supported by appropriate references.

: Thank you so much for the comment, we have seriously revised the whole sections of the manuscript particularly in interdiction, method and discussion according to the comments of reviewer 1 (including several excellent pieces comments presented in PDF file) and reviewer 2.

2. Please revise your "materials and methods" section to clarify your sampling frame, as well as inclusion and exclusion criteria. Also clarify issues of participant selection and methods applied to minimize bias in your study.

: Thank you, this section has been strongly reviewed and revised.

3. Please revise your "results" section to clarify major themes and subthemes. Also describe how data saturation was achieved, and revise Table 1 as recommended by reviewers to exclude/correct inappropriate columns and unnecessary inclusion within your Tables.

: We have re-analysis the whole data set and several pieces have been confirmed, while many other findings have been revised and improved including the correction of the number of “tribe (Akha and Lanu)” in table 1, thank you so much

4. Revise your discussion section with comparative analysis to similar studies in the literature, with appropriate referencing and how these other studies compare with findings from your own study.

: Thank you, this section has been largely revised and improved including moving into the proper place or alignment.

5. Clarify your Conclusions as well as your recommendations.

: We have revised the conclusion, but we would like to confirm the redocumentations under specific context of our location.

6. In addition, substantially revise your entire manuscript to address all grammatical and typographical errors. Consultation with an English language editing service is recommended to address this issue if necessary or if required by the authors.

: Basically, the manuscript has been improved by the AJE twice but definitely, the quality is still very poor. We are so sorry about this and have made our best capacity before submitting this version. If you have any better choice, please kindly let us know.

7. Kindly correct and edit to your COREQ checklist to include page numbers with reference to the manuscript.

: Thank you, it’s revised.

8. Please address all other relevant comments as annotated on the manuscript by reviewer #1 where necessary.

: Thank you, we have revised all points including comments presented in the PDF.

Additional requirements

: Thank you, we have added the correct information in the cover letter, please kindly see in cover letter.

: Thank you, this study was granted by the Center of Excellence for the Hill tribe Health Research, Mae Fah Luang University with the No. 1/2021.

: The funders had no role in study design, data collection and analysis, decision to publish, or preparation of the manuscript. Please kindly see in our cover letter.

: No. no researchers received a salary from the funder.

d) If you did not receive any funding for this study, please state: “The authors received no specific funding for this work.” Please include your amended statements within your cover letter; we will change the online submission form on your behalf.

: N/A

5. Thank you for stating the following in the Acknowledgments Section of your manuscript: "We thank Mae Fah Luang University and The Center of Excellence for The Hill Tribe Health Research for supporting the grants. " We note that you have provided additional information within the Acknowledgements Section that is not currently declared in your Funding Statement. Please note that funding information should not appear in the Acknowledgments section or other areas of your manuscript. We will only publish funding information present in the Funding Statement section of the online submission form.

Please remove any funding-related text from the manuscript and let us know how you would like to update your Funding Statement.

: Thank you, it’s removed.

Currently, your Funding Statement reads as follows:

: Thank you, we have mentioned all necessary in the cover letter, please kindly update the information in the system accordingly.

Reviewer #1: This topic would add knowledge to stigma-related studies. Nevertheless, the manuscript needs tremendous flow, coherency, and sentence structure improvement. The current writing style needs to be more scientific. A professional English editing service is deemed necessary.

1.The introduction did not clearly justify the rationale on the conduct of this study.

: We have seriously and heavily revised the whole section of the introduction. Thank you very much in all the comments along every sentence and paragraph.

2.The method need to be explained explicitly.

: Thank you, we have extended details as necessary, thank you so much.

3.The discussion is not adequate.

: Thank you, we have added as needed in this part, thank you.

4.The recommendation in the conclusion appeared abruptly.

: Thank you for the comment, we have seriously discussed in the recommendation, however with the context and the system (health system), we would like to maintain our original thought. However, we have revised the section of the conclusion.

Following are the comments and revision on PDF form

5. Introduction: overall, the introduction needed tremendous improvement in terms of presenting and justifying the facts. At the first sentence page 2, please rephrase this sentence. reviewer has difficulty in comprehending this sentence.

: There are rephrased, page 2 lines 24-25.

6. Please rephrase the second paragraph

: It is rephrased, page 2 lines 25-28.

7. Page 3 line 2-3, considering removing this sentence, just end the sentence with citation will do.

: Thank you, it’s improved accordingly (page 3, lines 2-4).

8. Page 3 line 3-4, please rephrase

: It’s rephrased.

9. Page 3, suggest stating the effect or giving example of these short and long term effect.

: It’s improved.

10. Page 3 the second paragraph, please cite references and please rephrase.

: Thank you, it’s cited.

11. Page 3, line 9, how many groups? and which group? Not clear to reviewer, please be specific.

: Since our original though, did not go to any specific group of people of the hill tribe. Rather identifying people who were living in specific geographical areas according to the inclusion and exclusion criteria, then we did not mention the specific group.

12. Page 3 line 10, unclear sentence. how often it was?

: It’s improved.

13. Page 3 line 11, what are the other characteristics? And how does the hill tribe people be related with the 'other characteristics' and causing them to be stigmatized? the inclusion of hill tribe people is abrupt in this sentence.

: It’s rewritten, lines 20-22.

14. Page 3, line 14, indicate or give example of the minority in Thailand.

: It’s improved

15. Page 3 line 16, which population? And what factors? please state or elaborate.

: It’s improved.

16. page 3 line 21, suggest omitting 'with ... digits'.

: Thank you, it’s improved accordingly.

17. page 3 line 22, please rephrase. Sentence structure error.

: Thank you, it’s rephrased.

18. page 3 line 24, please quantify or state what kind of healthcare problems?

: It’s added information.

19. page 3, line 25, please cite a reference and how does the underlying healthcare problems been associated with stigma? please justify

: It’s cited with justifications.

20. page 3 line 26-31, reviewer cannot connect the justification on why the author need to conduct this research. please define adaption at the prior paragraph or sentences. Sudden appearance of 'adaption' could put reader in wonder. what is the linkage of this sentence with the prior and latter sentence?

: Thank you very much, the paragraph and information have been reorganized and added information.

21. page 4 line 1-3, please rephrase this sentence.

: It’s improved

22. page 4 line 5-6, please define phenomenological method and justify the use of this method in your study and this sentence is ambiguous, not clear to reviewer.

: Since the terminology has its definition, we feel that it is not necessary to put any information in this sentence. We are sorry, we would like to maintain its original sentences in this context.

23. page 4 line 8-9, any reason selecting this age range? ,how did the author identified patients had experience of stigma? ,

: We use our inclusion and exclusion to screen those who met the criteria before the interview. To know about having stigma experience, they were explained and helped in identifying their experiences in facing the stigma.

24. And Also, kindly describe the tribes that is target population. A sudden appearance of different tribes at the results part may appear to confusing to some of readers.

: We have more than 15 years of experience in working research on the hill tribe people, then in practice, there were no mistakes. In simple response to the question, identifying the hill tribe we commonly used by asking them. We also do know them by language, dress, housing style, culture, etc.

24. page 4 line 11, please rephrase. And, what are the background of these key-informant?

: It’s rephrased with information.

25. page 4 line 19-20, unclear to reviewer. Please rephrase.

: It’s rephrased

26. page 4 line 25, do state the sampling frame. were these four villages resided by all hill tribes or these four out of XX number were selected due to the geographical location?

: There are 769 hill tribe villages located in Chiang Rai Province, but with the specific qualitative study design, we selected four villages that met the criteria (one located close to health promoting hospital, one located close to a district hospital, and two far away from hospitals and city). People who lived in these four villages and had stigma experienced were purposively selected.

27. page 4 line 30, what language medium was used for the interview session?

: We used Thai, and information was added.

28. page 5 line 3, what was the sampling technique? please describe.

: It’s added, purposively selection.

29. page 5 line 5-6, were each transcript analysed first before proceeding to interview the next participant? please elaborate. And any translation being done? who did the translation? what was the process?

: Since we could complete in a short period for the interviews, then the following processes were operated after the completion of the all interviews.

30. page 5 line 7, by whom? the interviewer?

: By researchers

31. page 5 line 15, do you mean 'those who are illiterate'?

: Who could not write and singe their name, it’s improved

32. page 5 line 18, do you mean audio records?

:Yes, it’s improved.

33. page 5 line 20, all translated quotes need improvement in sentence structure, grammar, etc...

: Thank you, it’s improved.

34. page 5 result, The themes and sub-themes generated were not concise. It appeared raw, and needed a lot of improvement. Please kindly do more literature search to derive more accurate theme and sub-themes. The theme 'responses' had a lot of sub-sub themes. This had confused the reviewers.

: Thank you, we revised the whole process of the analysis (this is the main reason for taking a long time to revise the manuscript). We found that we have made the proper and accurate findings. However, in the term “Response”, we changed to “Reaction to the experience” which is what we want to say.

35. page 5 line 24, please check if this is an appropriate way to report the min and max. and what is the unit, was this in percentage?

: Thank you, its’ revised and put in the proper place.

36. page 5 table 1, sex, suggestion: gender instead of sex

: Thank you, it’s changed.

37.page 5 table 1 age, any reason to categorise the age category as such?

: We used our experience base while conducting a quantitative study.

38. page 5 table 1, age , suggest presenting these number in the main text instead of adding into the table. please delete this row from the table

: Thank you, it’s improved

39.page 5 table 1, monthly income, present in the main text instead of table

: Thank you, it’s improved

40. page 7 line 10, kindly define form of stigma.

: Since it is in the result section, and it is widely used, including the surrounding context is saying what is the definition. Then we would like to maintain the original form. We do hope you understand.

41. page 7 line 16-17, please define verbal form of stigma. and cite reference for 'major form of stigma' and not clear to reviewer

: Since it is in the result section, and it is widely used, including the surrounding context

---

## [Decision Letter · Decision Letter 1]

20 Feb 2024

PONE-D-23-10436R1Stigma experiences and adaptations while accessing health care service among hill tribes in Thailand: a qualitative studyPLOS ONE

Dear Dr. Apidechkul, Thank you for submitting your manuscript to PLOS ONE. After careful consideration, we feel that it has merit but does not fully meet PLOS ONE’s publication criteria as it currently stands. Therefore, we invite you to submit a revised version of the manuscript that addresses the points raised during the review process.

 1. Kindly revise your Introduction section to provide a proper background on 'stigma" with appropriate references.2. Please thoroughly revise your Methods section to provide better clarification on 'inclusion and exclusion criteria';  more details on the qualitative analysis process, including how themes and subthemes where derived, methods for resolving intercoder agreements/disagreements, considering that the COREQ checklist mentions there were '21 data coders'?, etc.3. Consider rewriting the Results section based o the themes and themes derived from the study for better flow and clarity.4. Please submit an updated COREQ Checklist after the above revisions, especially with regards to the qualitative methods used in the study.5. Consider engaging the services of an English language editor to assist with language editing of the manuscript before resubmission.6. Kindly address all other issues raised by the peer-reviewers.

We look forward to receiving your revised manuscript.

Kind regards,

Sylvester Chidi Chima, M.D., L.L.M.

Academic Editor

PLOS ONE

Reviewers' comments:

Reviewer's Responses to Questions

**Comments to the Author**

1. If the authors have adequately addressed your comments raised in a previous round of review and you feel that this manuscript is now acceptable for publication, you may indicate that here to bypass the “Comments to the Author” section, enter your conflict of interest statement in the “Confidential to Editor” section, and submit your "Accept" recommendation.

Reviewer #1: All comments have been addressed

Reviewer #3: (No Response)

2. Is the manuscript technically sound, and do the data support the conclusions?

Reviewer #1: Yes

Reviewer #3: No

3. Has the statistical analysis been performed appropriately and rigorously?

Reviewer #1: N/A

Reviewer #3: N/A

4. Have the authors made all data underlying the findings in their manuscript fully available?

Reviewer #1: Yes

Reviewer #3: Yes

5. Is the manuscript presented in an intelligible fashion and written in standard English?

Reviewer #1: No

Reviewer #3: No

6. Review Comments to the Author

Reviewer #1: Overall comment: Though the manuscript has improved following round one revision, the writing skill can be improved further. Having professional English editing for academic papers would help tremendously. The current version is too lengthy, probably owing to limitations in writing or language proficiency. The manuscript could be more concise.

Reviewer #3: Reviewer comment on revision number 1:

I find the topic of the study interesting and commend the authors on being able to interview 85 individuals, which is a feat in qualitative studies. With 85 interviews of 45 minutes each, I was expecting much richer discussion and nuanced insights into the stigma experienced by the hill tribe people in Thailand. However, I find that there is a lack of clear understanding of the authors on stigma theory reflected through very little nuances and depth in the introduction, results, and discussion section of the paper. This is somewhat disappointing considering the time and effort of research participants and researchers that have gone into this. Unfortunately, I do not think that this manuscript is fit for publication in its current form in PLOS and would advise authors to consider doing more studies on stigma theories and bringing in more reflections on the results and discussion sections. See below some of my specific comments.

Introduction:

It seems that the authors do not have an in-depth understanding of stigma phenomena, and this is shown in the introduction section. The first sentence provides a definition that does not give a complete picture of stigma as ‘negative attitude’ is only one aspect of stigma. Their referencing of APA website rather than the most cited definitions in the stigma field provided by Goffman or Thornicroft is reflective of their lack of proper insight into the topic they have conducted the research on. Verbal and physical stigma are generally not the categorization of forms of stigma but manifestations of interpersonal stigma. It is recommended that authors conduct some review of stigma literature and theories to gain more insight into stigma processes.

Methods:

The lack of information in the methods section is concerning.

The authors mention that the inclusion criteria (pg.4, line4) were “hill tribe people aged >=15 years who had attended a hospital at least twice within the previous year and experienced stigma while accessing a service in a hospital. Those who had poor capacity to provide essential information according to the interview guide were excluded from the 7 study” but it is not clear what “poor capacity to provide essential information” means or who made the decision (and under what criteria) regarding the capacity.

Since this is also a qualitative study, the authors fail to reflect on what excluding people from a certain capacity to provide information would mean in terms of inclusivity of research and ethical implications around it.

The authors do not have information on the qualitative analysis process. Information is missing on what analysis method was used, how were the themes derived, how many people were involved in the coding process, how inter-coder agreement was assessed, etc.

Results:

In the form of stigma, the authors again talk about verbal, physical, and negative looks. These are examples of interpersonal stigma manifested as negative behavior by health workers towards hill tribe people. This interpersonal form of stigma is reported by the hill-tribe people and so can also be called experienced stigma. It would be better if the authors phrased it as the experienced stigma of hill tribe people while accessing healthcare services. Other forms of stigma that could have been explored by authors and would have been very interesting would have been self-stigma and structural stigma. In addition, the authors mention 4 forms of stigma were identified (pg.7 line 8) but describe only 3.

I find the quotes very interesting, but the entire result section lacks the depth and nuances of themes being discussed that a good qualitative study requires. It seems that the authors have just added quotes one after another without providing any breadth, depth, or nuances of the themes.

A similar issue is also seen in the discussion section where authors have just regurgitated the results.

7. PLOS authors have the option to publish the peer review history of their article (what does this mean? ). If published, this will include your full peer review and any attached files.

**Do you want your identity to be public for this peer review?** For information about this choice, including consent withdrawal, please see our Privacy Policy .

Reviewer #1: No

Reviewer #3: No

---

## [Author Response · Author response to Decision Letter 2]

22 Aug 2024

Respond to editor and reviewer’s comment

Thank you so much for the opportunity; we have improved all points of your concern. The manuscript has also improved its English by AJE, which is the second time. We do very much hope that you are happy with this version.

We invite you to submit a revised version of the manuscript that addresses the points raised during the review process.

1. Kindly revise your Introduction section to provide a proper background on 'stigma" with appropriate references.

Thank you. We have carefully revised the entire introduction section.

2. Please thoroughly revise your Methods section to provide better clarification on 'inclusion and exclusion criteria'; more details on the qualitative analysis process, including how themes and subthemes where derived, methods for resolving intercoder agreements/disagreements, considering that the COREQ checklist mentions there were '21 data coders'?, etc.

: Thank you for the comment, it’s revised and improved in the whole section

3. Consider rewriting the Results section based o the themes and themes derived from the study for better flow and clarity.

: This section is analyzed, and revised as suggested wholly.

4. Please submit an updated COREQ Checklist after the above revisions, especially with regards to the qualitative methods used in the study.

: It’s submitted together with this revised paper submission.

5. Consider engaging the services of an English language editor to assist with language editing of the manuscript before resubmission.

: The paper has been revised by native speakers.

6. Kindly address all other issues raised by the peer-reviewers.

: We have carefully revised and improved according to all the reviewers’ comments.

Reviewer #1: Overall comment: Though the manuscript has improved following round one revision, the writing skill can be improved further. Having professional English editing for academic papers would help tremendously. The current version is too lengthy, probably owing to limitations in writing or language proficiency. The manuscript could be more concise.

: Thank you for the comment. In this version, one more native English speaker helped us in editing the whole paper before submitting it.

Reviewer #3: Reviewer comment on revision number 1:

I find the topic of the study interesting and commend the authors on being able to interview 85 individuals, which is a feat in qualitative studies. With 85 interviews of 45 minutes each, I was expecting much richer discussion and nuanced insights into the stigma experienced by the hill tribe people in Thailand. However, I find that there is a lack of clear understanding of the authors on stigma theory reflected through very little nuances and depth in the introduction, results, and discussion section of the paper. This is somewhat disappointing considering the time and effort of research participants and researchers that have gone into this. Unfortunately, I do not think that this manuscript is fit for publication in its current form in PLOS and would advise authors to consider doing more studies on stigma theories and bringing in more reflections on the results and discussion sections. See below some of my specific comments.

Introduction:

It seems that the authors do not have an in-depth understanding of stigma phenomena, and this is shown in the introduction section. The first sentence provides a definition that does not give a complete picture of stigma as ‘negative attitude’ is only one aspect of stigma. Their referencing of APA website rather than the most cited definitions in the stigma field provided by Goffman or Thornicroft is reflective of their lack of proper insight into the topic they have conducted the research on. Verbal and physical stigma are generally not the categorization of forms of stigma but manifestations of interpersonal stigma. It is recommended that authors conduct some review of stigma literature and theories to gain more insight into stigma processes.

: Thank you so much for your kind comments. Even I have been trained in a two-week program under the FIC Stigma Research Training program (NIH) in 2021. I also attended several mental health, stigma, and qualitative classes while attending my coursework terms of my Master of Medical Science- Global Health Delivery (MMSC-GHD) at Harvard Medical School (cohort 2023-2025), I agreed with you that I need to learn more to understand the stigma which is very much interested to me.

Methods:

The lack of information in the methods section is concerning.

The authors mention that the inclusion criteria (pg.4, line4) were “hill tribe people aged >=15 years who had attended a hospital at least twice within the previous year and experienced stigma while accessing a service in a hospital. Those who had poor capacity to provide essential information according to the interview guide were excluded from the 7 study” but it is not clear what “poor capacity to provide essential information” means or who made the decision (and under what criteria) regarding the capacity. Since this is also a qualitative study, the authors fail to reflect on what excluding people from a certain capacity to provide information would mean in terms of inclusivity of research and ethical implications around it.

: Thank you so much for the great comment. We mentioned the exclusion in our proposal, and after careful discussion of the point, we felt that it was not a good step for the qualitative method, and we did not use it when selecting participants. We have deleted this sentence from the paper. We used a purposive method to select the participants with stigma experiences for the interviews.

The authors do not have information on the qualitative analysis process. Information is missing on what analysis method was used, how were the themes derived, how many people were involved in the coding process, how inter-coder agreement was assessed, etc.

: Thank you for the comment. We have extended the content in this section; Please see page 5, lines 1-8.

Results:

In the form of stigma, the authors again talk about verbal, physical, and negative looks. These are examples of interpersonal stigma manifested as negative behavior by health workers towards hill tribe people. This interpersonal form of stigma is reported by the hill-tribe people and so can also be called experienced stigma. It would be better if the authors phrased it as the experienced stigma of hill tribe people while accessing healthcare services. Other forms of stigma that could have been explored by authors and would have been very interesting would have been self-stigma and structural stigma. In addition, the authors mention 4 forms of stigma were identified (pg.7 line 8) but describe only 3.

: The section has been revised and improved.

: It has three forms, and it’s improved.

I find the quotes very interesting, but the entire result section lacks the depth and nuances of themes being discussed that a good qualitative study requires. It seems that the authors have just added quotes one after another without providing any breadth, depth, or nuances of the themes.

: We are so sorry to make you feel like that. We have revised the whole process of the analysis and revised some points that we could do. If you have any suggestions, please do so.

A similar issue is also seen in the discussion section where authors have just regurgitated the results.

: Thank you for the comment. We have revised many points in the discussion section, particularly your comments on the table. Thank you so much, and very appreciated.

1st revision: Stigma experiences and adaptations while accessing health care service among hill tribes in Thailand: a qualitative study

Overall comment: Though the manuscript has improved following round one revision, the writing skill can be improved further. Having professional English editing for academic papers would help tremendously. The current version is too lengthy, probably owing to limitations in writing or language proficiency. The manuscript could be more concise.

Section Context Reviewer’s Comment

Introduction Verbal stigma is the form of voice and language used for one’s negative attitude to the other [10]. Physical stigma could present in various forms to present one’s negative attitude to the other [10]. These two sentences can be merged.

: Revised, see lines 28-29, page 2

There is no evidence of stigma among the hill tribes while accessing health services in a health institute. Please rephrase this sentence

:Revised, see lines 20-21, page 3

- Overall, please improve introduction, justification on why the researcher need to determine stigma among hill tribes. Professional English editing will definitely help in improving the writing.

: All sections have been improved.

Result Negative looks can discourage people, making people think of themselves in negative ways, especially hill tribe people who live in poor communities in Thailand. This paragraph needed improvement.

: Revised, see lines 7-9, page 10

a. Completely not understood message, then no impact received Could be improved, needed to be more concise

: Revised, see line 13, page 10

Little understanding message, then little pain received Could be improved, needed to be more concise

: Revised, see line 10, page 11

Without effective communication, misunderstandings could impact several things. What kind of several things? Please describe.

: Revised, see line 16, page 11

c. Fully understood message, then with full impact received Could be improved, needed to be more concise

: Revised, see line 32, page 11

A nonresponse was one of the responses when the patient did not say anything. This sentence could be improved, try to rephrase it.

: Revised see lines 2-3 , page13

Responses The subtheme of non-responses could be merged under “responses”.

: Thank you for the comment. We, all authors, have discussed in this point seriously, and decided to maintain in current form to reduce the confusing. However, thank you so much for the suggestion.

- Overall, the results are clearly described, however, the sub-themes needed to be more concise.

There are too many sub-subthemes, which could be improved and merged further. The results could be consolidated further.

: Thank you for the comment. We took long time looking back to our raw data and repeating the process of the analysis, and at the end we would prefer to maintain the current form of our result presentation. If we merge any part, there will be impact on the raw data and the flow of getting the structure of our findings. We also have tried to give back the final form/ theme to the participants and found that it is perfectly accepted by the participants.

Discussion Those who did not understand the messages did not get the impact; those who had little understanding of the message got little pain, and those who fully understood the message got full impact. This sentence is redundant, please improve it.

: Revised, see lines 12-14, page 18

The hill tribe members had two main reactions when facing stigma: non-response and response (proper, nonproper immediate response, and aggressive response). A repetitive of the results.

: Revised, see line 14-15, page 18

Surprisingly, the ability to understand the message could lead to different impacts on the message receivers. If the situation of verbal stigma occurs, those who completely understand the message are fully impacted while those who do not understand the

message will have little or no impact. A repetitive of the results.

: Revised, see lines 19-20, page 18.

but the patients did not understand the messages received then these situations could develop intentional and unintentional actions which were one of the sources of stigma. Please cite a reference for this sentence

: Revised, see line 28 with the new reference no 29, page 18

that promoting a good organizational culture and developing healthcare workers’ interpersonal skills could minimize the impact of stigma on patients And hence, how this would imply to the findings of your study. What information or recommendations do you try to convey when basing this reference on your study?

: Thank you for the comment. This summarize comes from the findings presented that almost the stigma form hill tribe people received are contru0buted by healthcare workers behaviours.

Different levels of socioeconomic status……Livingston reported that stigma was a primary source

of mental health problems How did these two sentences be relevant to each other?

: Thank you for great comment. We agree with you that it is a new idea presented which is not clearly relevant to the key message of the paragraph, then we remove it.

.

These characteristics particularly protected the hill tribe people from the stigma encountered when attending a hospital. Redundant, it is repetitive in the previous sentence.

: Revised, see lines 25-27, page 19

- Overall, the discussion could be improved, need to have a more in-depth discussion.

: Thank you, we have carefully revised in the whole section including all points of your comments

Conclusion if they do fully understand, they will get full impact. Sentence hanging. The full impact of what? Please specify.

: Revised, line 10, page 20

Thank you so much.

TK

Assoc Prof. Dr. Tawatchai Apidechkul, MSc (Infectious Epidemiology), Dr. P. H (Epidemiology)

School of Health Science, MFU

Director, Center of Excellence of Hill Tribe Health Research

Former Hubert H Humphrey Fellow (2013-2014), Emory University

Global Health Delivery Intensive (Harvard School of Public Health)

Candidate MMSC-GHD (2025), Harvard Medical School, Harvard University

---

## [Decision Letter · Decision Letter 2]

2 Oct 2024

PONE-D-23-10436R2Stigma experiences and adaptations while accessing health care service among hill tribes in Thailand: a qualitative studyPLOS ONE

Dear Dr. Apidechkul,

Thank you for submitting your manuscript to PLOS ONE. After careful consideration, we feel that it has merit but does not fully meet PLOS ONE’s publication criteria as it currently stands. Therefore, we invite you to submit a revised version of the manuscript that addresses the points raised during the review process. **1. Kindly reevaluate and reconfigure the themes and sub-themes derived from your study after further in-depth analysis of your study data, to improve conciseness in characterization and reporting as suggested by Reviewer 1****2. Please address issues related to the abstract, discussion section and statement of imitations as suggested by Reviewer 4.****3. Consider engaging the services of an English language editor to improve the overall flow of your manuscript before resubmission.****4. Address al other issues raised by the peer reviewers.**

We look forward to receiving your revised manuscript.

Kind regards,

Sylvester Chidi Chima, M.D., L.L.M, LLD.

Academic Editor

PLOS ONE

Reviewers' comments:

Reviewer's Responses to Questions

**Comments to the Author**

1. If the authors have adequately addressed your comments raised in a previous round of review and you feel that this manuscript is now acceptable for publication, you may indicate that here to bypass the “Comments to the Author” section, enter your conflict of interest statement in the “Confidential to Editor” section, and submit your "Accept" recommendation.

Reviewer #1: All comments have been addressed

Reviewer #4: (No Response)

2. Is the manuscript technically sound, and do the data support the conclusions?

Reviewer #1: Partly

Reviewer #4: (No Response)

3. Has the statistical analysis been performed appropriately and rigorously?

Reviewer #1: N/A

Reviewer #4: (No Response)

4. Have the authors made all data underlying the findings in their manuscript fully available?

Reviewer #1: Yes

Reviewer #4: (No Response)

5. Is the manuscript presented in an intelligible fashion and written in standard English?

Reviewer #1: No

Reviewer #4: (No Response)

6. Review Comments to the Author

**Reviewer #1:**  Overall, I can tell that the authors had put great effort in this research work and manuscript. However, the works particularly the results section remains ‘raw’ and needed improvement. I understand that the current themes and sub-themes were the agreed upon among the authors, they lack element of conciseness. Considered read up more literatures for improvement of sentence structure, particularly the themes. Analysis of the data was deemed not in-depth. It can be delved further. An experienced senior qualitative researcher might help.

English writing skill in scientific paper needed improvement. A native speaker may not be sufficient in english editing of a scientific paper. A professional academic english editor may be necessary.

**Reviewer #4: ** Commendation on revision number 2

Dear Author,

Your manuscript contributes significantly to understanding the discrimination faced by Thailand's hill tribes in healthcare. It identifies the issues and proposes practical solutions. However, I've identified a few areas where improvements can be made to strengthen the paper further:

1. Abstract: In the end, you should restate the main points and highlight the steps that need to be taken. To improve the Hill tribe people, it must emphasize the importance of training healthcare workers, improving communication, and addressing socioeconomic inequalities.

3. Discussion: The demographic information could show why certain groups are important, like why it's interesting that many people don't have high education levels. This will define the context and highlight the significance of these characteristics in discussing stigma.

3. Limitations and Future Research: The study's weaknesses, including its small sample size or limited geographic focus, and suggestions for future research should be discussed in depth. This would strengthen the academic standard and underscore the need for more research on the topic, thereby motivating further investigation.

7. PLOS authors have the option to publish the peer review history of their article (what does this mean? ). If published, this will include your full peer review and any attached files.

**Do you want your identity to be public for this peer review?** For information about this choice, including consent withdrawal, please see our Privacy Policy .

Reviewer #1: No

Reviewer #4: No

---

## [Author Response · Author response to Decision Letter 3]

28 Oct 2024

Response to reviewers’ comments

Dear Editor and Reviewers,

Thank you so much for the excellent comments and the opportunity to revise this paper. We have carefully revised and improved based on all your comments. Two senior researchers have been asked to provide comments on the findings forming and English. Thank you, and I very much hope that you are happy with this version.

Reviewer #1: Overall, I can tell that the authors had put great effort in this research work and manuscript. However, the works particularly the results section remains ‘raw’ and needed improvement. I understand that the current themes and sub-themes were the agreed upon among the authors, they lack element of conciseness. Considered read up more literatures for improvement of sentence structure, particularly the themes. Analysis of the data was deemed not in-depth. It can be delved further. An experienced senior qualitative researcher might help.

: Thank you for your comment. We have revised the whole data analysis step again to make sure that we do not have any bias during the analyses and pick up only the evidence support without overclaimed interpretation before asking two senior qualitative researchers for their ideas. One of the critical points raised by our senior researchers is the paper is presented in a critical point with a simple presentation of the findings, which will make it easy for those policymakers in Thailand to further action implementation. We have been suggested to maintain the current form of the findings and share them with the policymakers working in the Ministry of Public Health, Thailand and also the WHO staff. They suggested that putting one more interpretation paragraph at the end of the results section would be great. We have put the final interpretation paragraph as suggested (page 18, lines 9-13), I do very much hope that you are happy and agree with us.

English writing skill in scientific paper needed improvement. A native speaker may not be sufficient in english editing of a scientific paper. A professional academic english editor may be necessary.

: Thank you. We have asked two senior researchers to modify the writing in a scientific style and made some modifications. They informed us that language is related to people’s cultures, and presenting based on their culture is a beautiful point of a qualitative study.

Reviewer #4: Commendation on revision number 2

Dear Author,

Your manuscript contributes significantly to understanding the discrimination faced by Thailand's hill tribes in healthcare. It identifies the issues and proposes practical solutions. However, I've identified a few areas where improvements can be made to strengthen the paper further:

1. Abstract: In the end, you should restate the main points and highlight the steps that need to be taken. To improve the Hill tribe people, it must emphasize the importance of training healthcare workers, improving communication, and addressing socioeconomic inequalities.

: Thank you for the comment; we have added the information in this section; please see page 2, lines 19-22.

3. Discussion: The demographic information could show why certain groups are important, like why it's interesting that many people don't have high education levels. This will define the context and highlight the significance of these characteristics in discussing stigma.

: Thank you for the critical point of comment. We have added information in page 18, lines 23-27.

3. Limitations and Future Research: The study's weaknesses, including its small sample size or limited geographic focus, and suggestions for future research should be discussed in depth. This would strengthen the academic standard and underscore the need for more research on the topic, thereby motivating further investigation.

: Thank you, we have added information on page 20, lines 15-17.

Thank you,

TK

TK

Assoc Prof. Dr. Tawatchai Apidechkul, MSc (Infectious Epidemiology), Dr. P. H (Epidemiology)

School of Health Science, MFU

Director, Center of Excellence of Hill Tribe Health Research

Former Hubert H Humphrey Fellow (2013-2014), Emory University

Global Health Delivery Intensive (Harvard School of Public Health)

Candidate MMSC-GHD (2025), Harvard Medical School, Harvard University

---

## [Decision Letter · Decision Letter 3]

19 Nov 2024

PONE-D-23-10436R3Stigma experiences and adaptations while accessing health care service among hill tribes in Thailand: a qualitative studyPLOS ONE

Dear Dr. Apidechkul,

Thank you for submitting your manuscript to PLOS ONE. After careful consideration, we feel that it has merit but does not fully meet PLOS ONE’s publication criteria as it currently stands. Therefore, we invite you to submit a revised version of the manuscript that addresses the points raised during the review process.

1. Kindly address any new issues raised by the peer reviewers.2.Please specifically address issues of English language editing, and the overall quality and standard of reporting the qualitative study within this manuscript.

We look forward to receiving your revised manuscript.

Kind regards,

Sylvester Chidi Chima, M.D., L.L.M, LLD.

Academic Editor

PLOS ONE

Reviewers' comments:

Reviewer's Responses to Questions

**Comments to the Author**

1. If the authors have adequately addressed your comments raised in a previous round of review and you feel that this manuscript is now acceptable for publication, you may indicate that here to bypass the “Comments to the Author” section, enter your conflict of interest statement in the “Confidential to Editor” section, and submit your "Accept" recommendation.

Reviewer #1: All comments have been addressed

Reviewer #4: All comments have been addressed

2. Is the manuscript technically sound, and do the data support the conclusions?

Reviewer #1: Partly

Reviewer #4: Yes

3. Has the statistical analysis been performed appropriately and rigorously?

Reviewer #1: N/A

Reviewer #4: N/A

4. Have the authors made all data underlying the findings in their manuscript fully available?

Reviewer #1: Yes

Reviewer #4: Yes

5. Is the manuscript presented in an intelligible fashion and written in standard English?

Reviewer #1: No

Reviewer #4: Yes

6. Review Comments to the Author

Reviewer #1: The study findings serve as valuable input for policy development. However, important aspects related to conducting qualitative research need to be addressed.

The quality of English writing in scientific papers remains unsatisfactory. It is recommended to use English editing and proofreading services

please see the attached word documents for more comments.

Reviewer #4: Dear author

The article is nearly ready for publication. The final suggestion is to discuss protective factors more. Being able to pay for speaking fluent Thai, wearing modern clothing, and paying medical fees to protect against stigma are significant. It would be better to compare it to other studies or discuss the possible social and economic impact.

7. PLOS authors have the option to publish the peer review history of their article (what does this mean? ). If published, this will include your full peer review and any attached files.

**Do you want your identity to be public for this peer review?** For information about this choice, including consent withdrawal, please see our Privacy Policy .

Reviewer #1: No

Reviewer #4: No

---

## [Author Response · Author response to Decision Letter 4]

16 Dec 2024

Stigma experiences and adaptations while accessing health care service among hill tribes in Thailand: A qualitative study

4th review

Overall comment The study findings serve as valuable input for policy development. However, important aspects related to conducting qualitative research need to be addressed

The quality of English writing in scientific papers remains unsatisfactory. It is recommended to use English editing and proofreading services

: It has been approved by AJE.

Study title Suggestion for improvement in sentence structure:

Stigma Experiences and Adaptations in Accessing Healthcare Services Among Hill Tribes in Thailand: A Qualitative Study

: Thank you, it has been revised.

Introduction No scientific evidence has been reported regarding the stigma experienced by hill tribes when accessing health services at healthcare institutions.

Comment: A study titled 'Components of Stigma and Its Impact on Maternal and Child Health Services and Outcomes: Perspective of Akha Hill Tribe Women in Thailand' was published in 2022. Please update your literature review accordingly

: Thank you, the sentence has been revised.

The author may need to include a definition of stigma adaptation in the introduction.

: Information has been added to page 3, lines 23–24.

Method Four villages were selected, but the approach used for sampling these villages needs to be specified. Was purposive sampling employed? What criteria were used to select these four villages?

: This has been clearly defined on page 4, lines 21–23.

Interview guide Improve this sentence: ‘The final question guide consisted of nine questions’

: We have revised the sentence; please see page 4, line 14.

Sampling method What method was used in sampling the participants?

: Purposive sampling was used; please see page 4, lines 23–24.

Data collection method All interviews were recorded after obtaining approval from the participant, and field notes were taken.

Comment: audio was recorded or audio-visual data was recorded? Please specify.

: Audiovisual recordings are not approved for use in any research. We referred to an audio recording, which is the process of capturing sound or speech using a recording device. This is clearly stated on page 5, line 3.

Sample size • I understand that the sample size in qualitative research is not pre-determined. However, I would like the authors to clarify the considerations used in estimating the number of participants to be included in the study.

: In the proposal development and IRB approval stage, we estimated that we needed 60 participants from those who had stigma experiences from accessing healthcare services. This has been added to page 4, line 23.

• Additionally, was there any estimation of the number of participants from each village?

: Yes, we estimated 15 people in each village were selected to be interviewed. This has been added to page 4, line 24.

• Were there any considerations regarding aspects such as age, gender, ethnicity, religions, incomes, or diseases when selecting participants, which might subsequently influence the sample size?

: No. At the beginning, we included the criteria of selecting villages and participants without any additional criteria regarding age, gender, ethnicity, religions, income, or diseases.

interviewer You have 11 interviewers. Did the author check interinterviewer reliability, or was any training provided to the interviewers for this study?

: Yes, all the interviewers had been trained in conducting in-depth interviews and had conducted least three in-depth interviews in project work experience. We all worked together during the proposal concept development, tool development and quality review, and cross-checked the completeness and quality during the whole process of data acquisition and analysis.

Saturation point What approaches were used to determine data saturation? Please specify.

: We went to the villages several times to complete the interviews with 60 participants. After the visits, we always discussed and recorded the key points. We determined data saturation had been reached when no more critical evidence, thoughts, experiences or possible themes were detected during our discussions.

Results "A certain level of improvement in the themes was noted."

The explanations of the sub-themes need improvement for clarity and accuracy, as there are difficulties in understanding the sentence structure. Professional English editing is recommended.

For example: ‘Proper response: In this type of response, the patient responds calmly and directly when a problem arises. Relatives initiated most responses; however, this situation indicates the lower 18 power of the responders to healthcare workers’, which can be improved to a better clarity paragraph:

"Proper response: In this type of response, the patient addresses problems calmly and directly when they arise. Although most responses were initiated by relatives, this indicates that patients often hold less power in interactions with healthcare workers."

: Thank you; we all agreed with you and replaced it.

Discussion ‘A large proportion of the hill tribe population has no formal education, which may result in limited access to health care services.’

Comment: How is the lack of formal education related to limited access to healthcare services? The author may want to clarify this connection.

: Thank you. We have improved this sentence; please see page 18, lines 24–25.

‘These basic profiles of the hill tribe people are working as the original sources of the stigma’

Comment: any literature to support this?

: This is the summary sentence which is essential to complete the expressions in the previous text.

‘To address the stigma issue among the hill tribe people, there should focus on the difference in stigma experienced in accessing different clinics.’

Comment: what does this means?

: It means that, in the interventions or implementations to address the stigma in these populations, the difference forms of the stigma exposed must be considered.

Future research or recommendation Should future research also investigate healthcare providers' perspectives toward the hill tribes?

: Thank you for excellent point; please see page 20, line 16.

limitation Were only literate participants included in the study? If so, this should be highlighted as a limitation, as it may have excluded the perspectives of non-literate individuals who could offer valuable insights.

: No, we did not focus on only literate participants, but rather all people who lived in the selected villages and had stigma experiences.

---

## [Decision Letter · Decision Letter 4]

21 Jan 2025

PONE-D-23-10436R4Stigma experiences and adaptations in accessing healthcare services among hill tribes in Thailand: A qualitative studyPLOS ONE

Dear Dr. Apidechkul,

Thank you for submitting your manuscript to PLOS ONE. After careful consideration, we feel that it has merit but does not fully meet PLOS ONE’s publication criteria as it currently stands. Therefore, we invite you to submit a revised version of the manuscript that addresses the points raised during the review process.

1. Please address all new issues raised by Reviewer 1 as annotated in your latest rebuttal letter.2. Kindly review any other English language issues in your revised manuscript before resubmission. Please submit your revised manuscript by Mar 07 2025 11:59PM. If you will need more time than this to complete your revisions, please reply to this message or contact the journal office at plosone@plos.org . Please include the following items when submitting your revised manuscript:

We look forward to receiving your revised manuscript.

Kind regards,

Sylvester Chidi Chima, M.D., L.L.M.

Academic Editor

PLOS ONE

Journal Requirements:

Reviewers' comments:

Reviewer's Responses to Questions

**Comments to the Author**

1. If the authors have adequately addressed your comments raised in a previous round of review and you feel that this manuscript is now acceptable for publication, you may indicate that here to bypass the “Comments to the Author” section, enter your conflict of interest statement in the “Confidential to Editor” section, and submit your "Accept" recommendation.

Reviewer #1: All comments have been addressed

2. Is the manuscript technically sound, and do the data support the conclusions?

Reviewer #1: Yes

3. Has the statistical analysis been performed appropriately and rigorously?

Reviewer #1: N/A

4. Have the authors made all data underlying the findings in their manuscript fully available?

Reviewer #1: Yes

5. Is the manuscript presented in an intelligible fashion and written in standard English?

Reviewer #1: Yes

6. Review Comments to the Author

Reviewer #1: The majority of the comments have been addressed; however, some minor touch-ups would help improve the overall presentation of this manuscript.

7. PLOS authors have the option to publish the peer review history of their article (what does this mean? ). If published, this will include your full peer review and any attached files.

**Do you want your identity to be public for this peer review?** For information about this choice, including consent withdrawal, please see our Privacy Policy .

Reviewer #1: No

---

## [Author Response · Author response to Decision Letter 5]

15 Feb 2025

Stigma experiences and adaptations while accessing health care service among hill tribes in Thailand: a qualitative study

5th review

Overall comment There were noticeable improvements in the writing and sentence flow after utilizing English editing services. However, the quality of English writing in scientific papers still leaves room for improvement.

Study title Query dissolved.

Introduction A few studies have reported on the stigma experienced and stigma adaptation among the hill tribes while accessing health services in at a health institute.

Comment: Please cite this statement when you mentioned ‘a few studies.’

: References 20 is placed

A few studies have reported on the stigma experienced and stigma adaptation among the hill tribes while accessing health services in at a health institute.

Comment: please state or describe the findings of these studies specially about stigma adaptation.

: Accepting the situation and choosing a private clinic were their adaptations (see page 3, lines 23-24)

Sample size • I understand that the sample size in qualitative research is not pre-determined. However, I would like the authors to clarify the considerations used in estimating the number of participants to be included in the study.

In the proposal development and IRB approval stage, we estimated

that we needed 60 participants from those who had stigma

experiences from accessing healthcare services. This has been added

to page 4, line 23.

Comment: Please read about sample size estimation in qualitative studies. Typically, at least six participants are required for an interview topic, but the ultimate sample size depends on the complexity of the topic. Some references suggest sampling 10 to 15 participants as an estimate for a proposal. Therefore, I assume your 60 samples are based on 15 participants per village. Please use this rationale and cite an appropriate reference to support it.

• Additionally, was there any estimation of the number of participants from each village?

Yes, we estimated 15 people in each village were selected to be

interviewed. This has been added to page 4, line 24.

Comment: same as above

:Thank you, we have added information with reference in page 4, lines 26-27.

interviewer You have 11 interviewers. Did the author check inter-interviewer reliability, or was any training provided to the interviewers for this study?

Yes, all the interviewers had been trained in conducting indepth

interviews and had conducted least three in-depth

interviews in project work experience. We all worked together

during the proposal concept development, tool development and

quality review, and cross-checked the completeness and quality

during the whole process of data acquisition and analysis.

Comment: please state this sentence in the manuscript ‘all the interviewers had been trained in conducting in-depth interviews and had conducted least three in-depth interviews in project work experience.’

: Thank you, it’s added in page 5, lines 5-6.

Saturation point What approaches were used to determine data saturation? Please specify.

: We went to the villages several times to complete the

interviews with 60 participants. After the visits, we always

discussed and recorded the key points. We determined data

saturation had been reached when no more critical evidence,

thoughts, experiences or possible themes were detected during our discussions.

Comment: please include this statement in the manuscript

: Thank you it’s added in page 5, lines 12-15.

Discussion

‘These basic profiles of the hill tribe people are working as the original sources of the stigma’

Comment: any literature to support this?

This is the summary sentence which is essential to complete the

expressions in the previous text.

Comment: This is a conclusive statement, but it is not based on your own study findings unless you cite literature to support the claim that isolated geographical location, low health literacy, and/or lower socio-economic status are the primary sources of stigma. Alternatively, you could rephrase your statement to better summarize your study findings.

: Thank you, it’s revised, please see page 19, lines 8-9.

‘To address the stigma issue among the hill tribe people, there should focus on the difference in stigma experienced in accessing different clinics.’

Comment: what does this means?

It means that, in the interventions or implementations to address

the stigma in these populations, the difference forms of the stigma

exposed must be considered.

Comment: Please include the rephrased sentence in place of your original one.

: Thank you, please see page 20, lines 29-30.

---

## [Decision Letter · Decision Letter 5]

3 Mar 2025

Stigma experiences and adaptations in accessing healthcare services among hill tribes in Thailand: A qualitative study

PONE-D-23-10436R5

Dear Dr. Apidechkul,

We’re pleased to inform you that your manuscript has been judged scientifically suitable for publication and will be formally accepted for publication once it meets all outstanding technical requirements.

Kind regards,

Sylvester Chidi Chima, M.D., L.L.M, LLD.

Academic Editor

PLOS ONE

Reviewers' comments:

Reviewer's Responses to Questions

**Comments to the Author**

1. If the authors have adequately addressed your comments raised in a previous round of review and you feel that this manuscript is now acceptable for publication, you may indicate that here to bypass the “Comments to the Author” section, enter your conflict of interest statement in the “Confidential to Editor” section, and submit your "Accept" recommendation.

Reviewer #1: All comments have been addressed

2. Is the manuscript technically sound, and do the data support the conclusions?

Reviewer #1: Yes

3. Has the statistical analysis been performed appropriately and rigorously?

Reviewer #1: N/A

4. Have the authors made all data underlying the findings in their manuscript fully available?

Reviewer #1: Yes

5. Is the manuscript presented in an intelligible fashion and written in standard English?

Reviewer #1: Yes

6. Review Comments to the Author

Reviewer #1: I believe that there is room for improvement for this paper in qualitative study conduct. Do continue seeking improvement in qualitative study and not be limited to the current approach of analyzing the qualitative data. The approaches the author uses in qualitative data analysis remain green and superficial, not in-depth.

Although the current work is below the standard of qualitative study presentation, the English writing skill is not optimal in presenting scientific works. The flow is not easy to read, and the sentences were not concise.

However, I am willing to give an 'acception' to the current work given the importance of knowing the stigmatization experienced by the minority population.

However, the final decision of acceptance remains with the editor.

7. PLOS authors have the option to publish the peer review history of their article (what does this mean? ). If published, this will include your full peer review and any attached files.

**Do you want your identity to be public for this peer review?** For information about this choice, including consent withdrawal, please see our Privacy Policy .

Reviewer #1: No

---

## [Editor Report · Acceptance letter]

PONE-D-23-10436R5

PLOS ONE

Dear Dr. Apidechkul,

I'm pleased to inform you that your manuscript has been deemed suitable for publication in PLOS ONE. Congratulations! Your manuscript is now being handed over to our production team.

Kind regards,

on behalf of

Professor Sylvester Chidi Chima

Academic Editor

PLOS ONE